# A Non-Invasive Sound Technology to Monitor Rumen Contractions

**DOI:** 10.3390/ani12172164

**Published:** 2022-08-24

**Authors:** Einar Vargas-Bello-Pérez, André Luis Alves Neves, Adrian Harrison

**Affiliations:** 1Department of Animal Sciences, School of Agriculture, Policy and Development, University of Reading, P.O. Box 237, Earley Gate, Reading RG6 6EU, UK; 2Production, Nutrition and Health, Department of Veterinary and Animal Sciences, University of Copenhagen, Grønnegårdsvej 3, 1870 Frederiksberg C, Denmark; 3PAS, Section for Physiology, Department for Veterinary and Animal Sciences (IVH), Faculty of Health & Medical Sciences, University of Copenhagen, Dyrlægevej 100, 1870 Frederiksberg C, Denmark

**Keywords:** cows, acoustic, rumen sounds, enteric gas, contractions, rumen movements

## Abstract

**Simple Summary:**

The recording of rumen sounds can be an important tool to assess the health status of a herd, as the frequency and amplitude of rumen contractions are associated with the flow and fermentation patterns of feed particles in the ruminoreticulum compartment. In this technical report, using a wireless device (CURO MkII), we recorded high-quality rumen sound waves that differentiate rumen contractions between cows of different production statuses (dry cow vs. lactating cow) and physiological stages (pregnant vs. non-pregnant). We envision that the rumen sound recordings will be a new form of technology to detect the onset of metabolic diseases, such as acidosis and hypocalcemia, which usually cause changes in the physicochemical properties of the rumen environment and reduce the frequency of rumen contractions. However, this technology still needs further improvement and validation through recordings of rumen movements in animals experiencing metabolic diseases, which will enable early detection of the problem and reduce treatment costs and production losses.

**Abstract:**

This technical report used a wireless device (CURO MkII) that recorded high-quality rumen sound waves from cows of different production statuses (dry cow vs. lactating cow) and physiological stages (pregnant vs. non-pregnant). Recordings from a dry Jersey heifer fed a diet based on haylage and straw showed a few high-amplitude spikes (3 at 6 dB) but mostly infrequent signals (9 at 12 dB and 22 at 18 dB), with pauses of approx. 2 min with no rumen sounds in between. Analysis of a few individual spikes in the 12 dB range showed that wave frequencies ranged from 230 to 250 Hz and lasted 4 s. Recordings of the high-yielding Red Danish cow fed a total mixed ration (TMR) showed an almost constant frequency of the rumen sounds with considerable amplitude of the waves. Rumen sounds from the Red Danish dry and pregnant cow fed on TMR were less frequent, with a lower amplitude than those from the high-yielding cow. These preliminary results demonstrate that wireless sound recording units are capable of measuring rumen sounds in a production setting and can discern between animals of different production and physiological stages, but more studies are needed to confirm our findings.

## 1. Introduction

The rumen is the largest compartment of the ruminant gastrointestinal tract and the main site of feed fermentation [1]. The rumen functions as an anaerobic chamber that provides the ideal temperature (38–42 °C) and pH (5.5–7.0) for the growth of a vast array of anaerobic and facultative anaerobic microbes [2]. The rumen has pillars or baffles, and its contraction forces the digesta back and forth across these structures in mixing movements. Mixing movements inoculate the ingested feed with the microbes and transfer fermentation acids arising from microbial fermentation to the epithelial surface so they can be absorbed. Pressure in the rumen triggers the extension of the cranial pillar, holding the digesta away from the esophagus, and facilitates the release of fermentation gasses by eructation. The rumen movements are coordinated by the activity of the vagus nerve (the 10th cranial nerve), which is activated by the tactile stimulation of feed particles present in the rumen. The reticulum is a small but distinct portion of the rumen that plays a key role in both rumination and digesta flow to the lower gut. Small particles are transferred from the rumen and reticulum to the omasum, but large and potentially digestible feed particles are forced back up the esophagus to the mouth for further particle size reductions in a process called rumination. Re-chewing the cud (rumination) increases the surface area of feed particles, fermentation rate, and the extent of digestion [3].

The duration and frequency of rumination, and the subsequent digesta flow to the lower gut, is affected by the type of feeding (concentrate vs. forage-based diets) and the physiological and health status of the animal [1]. Thus, the precise recording of rumen sounds could provide clues on the dynamics of the ruminal movements and the health of the animal, but implementing such technology poses technical challenges and requires monitoring systems that provide real-time data generated from the rumens of cattle raised on-farm.

Rumen function in dairy cows is essential to digestive efficiency, milk production, and farm profitability [4]. Usually, rumen function is evaluated by examining rumen motility, which is controlled by the parasympathetic nervous system. This examination is normally performed by veterinarians on-farm; however, it is labor-intensive, costly for farmers, and is only requested for cows exhibiting clinical signs of diseases [5].

Rumen sounds are important for health assessment as the frequency and amplitude of rumen contractions are influenced by the fermentation patterns of feed particles, and abnormal movements could be related to metabolic diseases such as acidosis and hypocalcemia [6]. At present, there are several technologies to monitor rumen function but most of them rely on the use of a bolus that demands trained personnel, animal stress, and time-consuming tasks to fetch the animals and administer the bolus. The use of a modern rumen sound technology is a non-invasive procedure that could considerably reduce animal stress and the costs associated with the health assessment of the herd. Earlier, Wynn and colleagues [7] were able to record sound waves of ruminal contractions by standing in person alongside a ruminant with a stethoscope connected to a sound recorder. Wynn and colleagues [7] indicated ruminal contractions from a recording made most likely before feeding, if not whilst the animal was being fed. In the present technical note, we monitored rumen sounds, recording those associated with A- and B-wave muscular contractions, those associated with digesta interactions and the rumen compartments, and those involved with the removal of fermentation products from the ventral sac, the ventral blind sac, and the dorsal blind sac.

## 2. Materials and Methods

### 2.1. Ethics

Animals used in this study were included in a license approved by the Department of Veterinary and Animal Sciences from the University of Copenhagen (protocol ID 2018-15-0201-01462).

### 2.2. Cows

A Jersey heifer (Cow #1) weighing 400 kg was used for recordings of basal activity at the Large Animal Teaching Hospital (Taastrup) from Copenhagen University. This animal was not pregnant, and had never been so, and was therefore fed a maintenance diet based on ad libitum meadow haylage (50%) and straw (50%), with 3 kg of concentrate per day. Two additional adult Red Danish cows weighing 600 kg were subsequently measured at Assendrup Hovedgård (Haslev, Denmark): (1) a high yielding cow with 100 days of lactation (Cow #2), and (1), a dry and pregnant cow (Cow #3). These two animals received a total mixed ration (TMR) ad libitum based on maize and grass silages supplemented by concentrate from milking robots (Lely, Maassluis, the Netherlands) according to their daily milk yield. The diet was composed of grass silage (43.6% DM), maize silage (36.2% DM), rapeseed cake (10.1% DM), wheat grain (5.5% DM), wheat straw (2.6% DM), and vitamin and minerals premix (2% DM). Animals were fed to fulfill the nutritional requirements for lactating cows based on a daily consumption of 21.4 kg of DM, and for dry cows based on 9.1 kg of DM, according to the Nordic Feed Evaluation System [8].

### 2.3. Rumen Sounds

The sound recordings were made using a CURO MkII unit (CURO-Diagnostics ApS, Bagsværd, Denmark) (Figure 1 and Figure 2). Sound recordings lasted 20 min for each animal and were performed 2 h after the morning meal. Recordings were performed in an open space on-farm. The recording site was prepared with acoustic gel (CURO-Diagnostics ApS), which was thoroughly rubbed into the overlying hair to ensure a good connection with the skin above the rumen, located on the left side of the animal in between the last rib, paralumbar fossa, and lumbar vertebrae. Sensors (piezo ceramic—diameter of 50 mm) were likewise prepared with acoustic gel, before they were attached to the site of interest using flexible self-adhesive bandage (Animal Polster, Snögg Industry AS, Kristiansand, Norway) (Figure 2). The sensors were then connected to the CURO MkII unit and recordings were made in the form of a WAV file to an iPad via a wireless connection (Figure 2), so recordings could be monitored in real-time. Data collection used a sampling frequency of 2000 Hz.

The CURO MkII units weighed 10.2 g and had a Bluetooth 4 connection to both tablet/smartphone-supported software. It had a battery life of 6 h non-stop recording. The range on the Bluetooth connection was approx. 40 m, but through a remote link it was also possible to place a CURO MkII unit on a cow, record rumen sounds from many kms away, and transmit them to a Client Cloud where they could be accessed from anywhere around the world.

### 2.4. Analysis

The raw WAV file recordings were analyzed using the CURO-Diagnostics software (https://app.myodynamik.com; accessed on 9 November 2021) and WavePad (WavePad Master Edition version 10.89—NCH Software—https://www.nch.com.au/wavepad/index.html (accessed on 9 November 2021); Greenwood Village, CO 80111, USA) in terms of a temporal frequency analysis (Figure 3). No processing of the data was carried out, but the recorded signal was monitored in real-time at the time of recording to make sure that sensor placement and gain setting were optimal.

## 3. Results

### 3.1. Proof of Concept

The equipment used for this trial proved successful in being able to measure rumen sounds and relay them via a Bluetooth 4 link to an iPad, where they could be viewed in real time up to approx. 40 m away. Moreover, the CURO Mk II units were so light that they did not affect the animal in any way. Taped to the abdomen of the animal, the piezo ceramic sensors coupled to the CURO units remained firmly in place and were not affected by tail movements or occasional licks from the subject.

### 3.2. Cow #1

The recordings from this subject represent those of a dry cow fed a maintenance diet. This recording shows a few high amplitude spikes (3 at 6 dB) but mostly infrequent signals (9 at 12 dB and 22 at 18 dB) with long pauses of approx. 2 min with no rumen sounds in between. Moreover, analysis of a couple of the individual spikes in the 12 dB range show them to have a frequency range of 230–250 Hz and a duration of 4 s (Figure 1).

Signal processing, and in particular the use of time-frequency analyses, are techniques that enable users to study a signal in both the time and frequency domains simultaneously. The motivation for such an analysis is that functions and their transforms are often tightly connected, and to understand them better they are often studied jointly. In this study, we used a temporal frequency analysis of rumen sounds to verify whether signal identification was at all possible. Note that despite the similar frequency ranges for the two rumen sound recordings (234–249 Hz), their amplitude was slightly different—52 dB left, and 47 dB right (Figure 3). These findings suggest that the quality of the obtained rumen sounds lends itself to the automated identification of specific rumen events and, with it, improvements in Precision Livestock Farming.

### 3.3. Cow #2

The recordings from this subject represent those of a high-yielding animal fed accordingly. Note not only the almost constant frequency of the rumen sounds, but also the considerable amplitude of these sounds (Figure 4—upper panel).

### 3.4. Cow #3

The recordings from this subject represent those of a dry cow. Rumen sounds were less frequent, with a lower amplitude, than those from the high-yielding cow (Figure 4—lower panel).

## 4. Discussion

The use of sensors is part of a technological approach known as Precision Livestock Farming. They are considered wearable devices that are wireless, small, and compact, and are resistant to the farm environment and temperatures and can be used to transmit high-speed information [9]. To date, sensors designed for ruminants can be found as a bolus, ear tags, collar or pedometers, and they can be used to monitor temperature, rumen pH, pressure, activity, or heart sounds [9]. However, the nature of the sensors described in this note has not been reported before. The information transmitted through the devices described in this note can be monitored remotely, and the portability of the sensors is optimal for farm animals. In this technical note, the advantage of using the described sensor is not only their size and weight but also their signal range and the ability to monitor animals in real-time. It is envisaged that veterinarians could use such a device to monitor rumen function in animals that have been affected in some way, enabling them to monitor the effects of treatment and the recovery process without having to be alongside them.

To the best of the authors knowledge, this is the first wireless measurement of rumen sounds made in real-time that is capable of detecting differences in individual animals that belong to different production levels, as shown by the distinct patterns of sound amplitude and frequency. A research group from Japan [6] reported a sensor that monitors ruminal motility, but this used a bolus-type wireless sensor that needs to be administrated orally using a catheter into the rumen; therefore, it requires trained personnel and time-consuming tasks to fetch animals that can lead to stressful conditions. For movement detection related to chewing–rumination monitoring, other sensor systems have used an accelerometer mounted on an ear tag (e.g., CowManager SensOor, Agis Automatisering BV, Harmelen, the Netherlands; Smartbow, Smartbow GmbH, Jutodasse, Austria) or neck collar (e.g., MooMonitor+, DairyMaster, Causeway, County Kerry, Ireland; eating only: CowScout Neck, GEA, Zurich, Switzerland).

Data from Cow 1 belong to a non-pregnant, non-lactating Jersey heifer that was fed on hay, while Cow 3 data are from a Red Danish dry pregnant cow fed with a total mixed ration. In an earlier report, Welch [10] discussed the differences in rumination times between dairy cattle breeds (Guernsey, Ayrshire, Holstein, and Jersey), as their body size and weight is reflected in their rumen wall size and volume capacity. In this report, we saw differences between dry cows that might be attributed to the reproductive status of the animal (pregnant vs. non-pregnant). It is worth noting that a growing fetus will reduce the space inside the abdomen, reducing the space for the rumen to move. Therefore, the dry matter intake reduces drastically as the calving date approaches [11].

Differences between rumination times using the Hi-Tag acoustic system have been highly correlated with direct observations [12] and video recordings [13]. However, as observed in this note, there is substantial variation among individual cows. Researchers have attributed such differences to neck muscle or skin thickness that interferes with the placement and function of the sensor, which could produce background sounds [14]. In any case, monitoring several minutes per day when cows are eating and ruminating is critical to note subtle changes in their behavior, enabling the early identification of cows that are overfed, underfed, sick, or even experiencing estrus, rumen acidosis or hypocalcemia.

The lactation stage leads to physiological changes that also affect rumen motility. Differences between Cow 2 vs. 1 and 3 are clearly due to the amount of feed intake, as well as the productive stage. This has been assessed by a Dutch research team [15] using a three-dimensional (3D) vision-based system showing rumen fill changes over days in milk. The Dutch system can even detect rumen dysfunction associated with a decrease in milk production 4 days before the farmer’s detection. In this case, if our sensor recordings are matched with milk yield and milk composition, we could be able to detect rumen motility problems at an early stage. However, further data processing is required to establish “normal” parameters, and record rumen sounds at a herd level over a complete lactating period. Taken together, preliminary results from Cow 1 (non-pregnant, non-lactating Jersey heifer), 2 (Red Danish lactating cow fed on TMR) and 3 (Red Danish dry and pregnant cow fed on TMR) showed that diet, breed, age, and body mass affect rumen sound and further studies are needed to confirm our findings.

In this technical note, the data presented from a simultaneous recording of two Red Danish cows, one dry and one that was 100 days into lactation (Figure 4), clearly illustrate not only the clear difference in rumen sounds that occurs at production level, but also the ease with which such animals can be monitored. When assessing ruminants in a field or production setting, rumen sound recordings obtained in this way can add important additional information that cannot otherwise be obtained in a non-invasive fashion. Such data may be used to make ruminant production efficiency improvements as well as provide critical and much-needed information about eructation and its role in global warming.

The authors envisage the combined use of a CURO MkII unit alongside measurements of ruminant gas production not only as a means of identifying animals with a high gas production, but also as a means of monitoring the efficacy of compounds aimed at reducing ruminant gas production globally. However, obtained recordings now need to be validated with measurements of rumen fermentation parameters at an in vivo level, with animals at different production or physiological stages. In the mid-term, the use of this technology could be used to monitor rumen function and health and help with the early detection of digestive disorders.

## 5. Conclusions

Overall, wireless sound recording units, like the CURO MkII, are capable of measuring rumen sounds in a production setting and can discern between individual animals of different production levels. The results showed that diet, breed, age, and body mass affect rumen sounds.

At this stage, data integration will be the next step for the use of the CURO MkII. Data need to be processed using a static algorithm, artificial intelligence, or machine learning [16]. This future application could predict a cow’s status based on her previous data recording and generate a forecast of the desired status for an animal at a given stage of the production cycle [17]. It is important to note that this is a very preliminary report and further studies are needed to confirm our findings.

## Figures and Tables

**Figure 1 animals-12-02164-f001:**
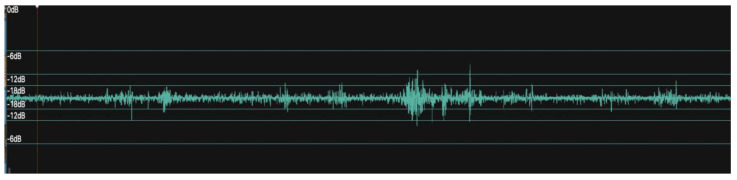
An example of our own rumen sound recording (Cow #1) made using a CURO MkII wireless system (CURO-Diagnostics ApS, Bagsværd, Denmark) operating Bluetooth 4 and interacting with a recording device remotely. Sampling rate 2 kbps, gain 32 dB. This dataset represents short samples of a longer 34-min recording. Rumen sounds were measured on the decibel (dB) scale. Note the occasional bursts of rumen sounds with relatively long periods of little to no rumen sounds in between them.

**Figure 2 animals-12-02164-f002:**
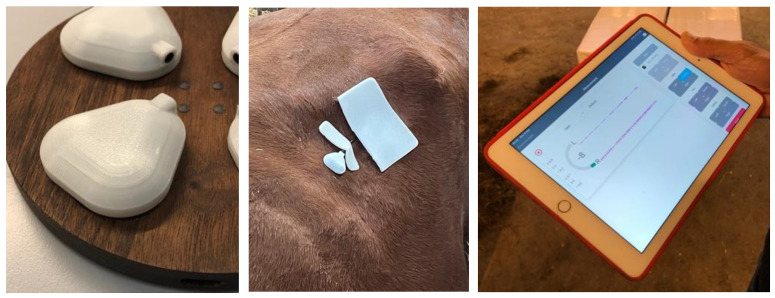
A CURO sensor unit on its charging plate (**left**), positioned on the ruminant at the site of interest (**middle**), generating recordings of rumen sounds in real time to an iPad (**right**).

**Figure 3 animals-12-02164-f003:**
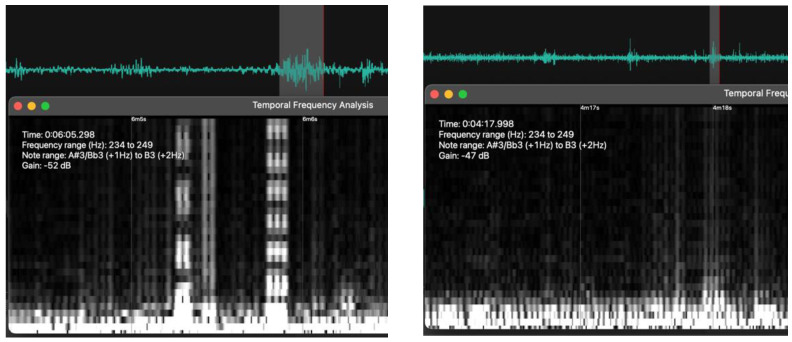
A temporal frequency analysis of the rumen sound signals showing two spikes highlighted (grey shade) and analyzed for their frequency range. Analysis performed using WavePad.

**Figure 4 animals-12-02164-f004:**
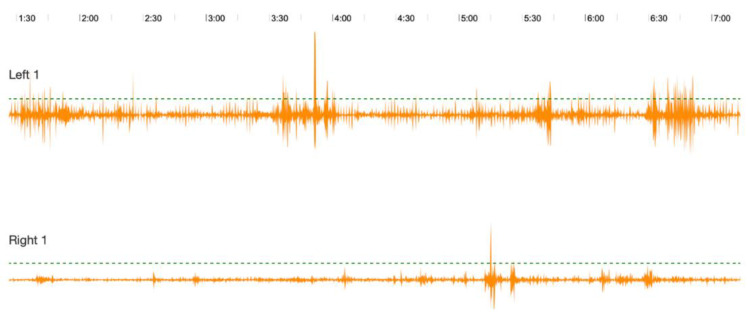
Rumen sound recordings made from a CURO MkII simultaneously from a lactating (Cow #2; Left 1) and a dry cow (Cow #3; Right 1), showing the increased amplitude and frequency of rumen sounds in the lactating cow. Recordings were taken from animals from Assendrup Hovedgård (Haslev, Denmark). The sound recording in the lower panel Cow #3 is very similar in amplitude and frequency to those sounds recorded for Cow #1 (Figure 1). The number scale at the top represents time in minutes.

## Data Availability

The datasets used in this study can be provided for free use upon request from the corresponding author.

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
