# Peer review of "A Non-Invasive Sound Technology to Monitor Rumen Contractions"

_animals, 2022, doi:10.3390/ani12172164_

Round 1

Reviewer 1 Report

Animals 1837302

A non invasive sound technology to monitor rumen contractions

This is a well-written technical note, only a few recommendations are made.

The biggest concern is the lack of sample size (3) with no replication which is noted as a limiting factor. It is noted that the Curo MkII was used only once per cow for 20 min (L111-112), is this correct? Was there a reason why this was not extrapolated out for a longer period? I also understand that this is potentially commercially sensitive data (L289-291) and there was no funding provided (L280). This manuscript can be recommended for publication as it is a technical note and is novel enough for consideration in spite of the lack of sample size.

L250 states that the authors aim to move to an entire herd over a complete lactating period – this is a big jump from the present study which is only 3 cows for a total of 20 minutes

A bit confused also of the range – it says 40 metres for Bluetooth but then “anywhere around the world”, what is the remote link and how does it work?

Minor comments:

L61-62 – a bit confused by this, the technology has the objective of automation and provide clues on ruminal movement dynamics but there is a challenge of monitoring systems? The monitoring systems should be an advantage, what is the challenge to obtain this real-time data?

L79 – change “To this end” to “In the present technical note,”

L90 – how old was cow #1, was she a heifer?

L97 – change “dry” to “dry and pregnant” as per L225. Add “These two” before animals

L100 – put grass silage before maize silage and put space between % and DM

L101 – put space between % and DM

L103 – after “kg of DM”, add “, and for dry cows based on 9.1 kg,”

L104 – remove the final sentence

L106 – add “the” before solid black

L128 – no need for capital w for world

Figure 3 – put the right panel in the middle as it flows better

L155 – change “subject” to “animal”

L157 – were these licks and tail movements observed during the 20 minutes of experimentation?

L170 – put comma after (234-239 Hz)

L173 – change “with-it” to “with it,”

L177-180 – these sentences starting from “Note that” can be removed as they were mentioned in the text

L184 - change “trace” to “panel”

L188 - change “trace” to “panel”

Figure 5 – it may benefit to label the two panels as “Cow #2” (Left 1) and “Cow #3” (Right 1), the left and right names are a bit confusing

L222 – change “Co.” to “County”

L224 – is this paragraph comparing two non-lactating cows from different breeds? It may be worth mentioning Cow 3’s diet

L252 – change “a dry cow” to “two Red Danish cows, one dry” as this compares the same breed at two different stages of lactation

L265 – put in vivo in italics

L266 – put “and” between function and health and add comma after health

L269 – remove “the”

References – only half of these show DOI and there are differences in “volume: pages” or “volume, pages”

Reference 2 – remove “New York State College Of, A., Life, S. & Department of M.” and move “Ithaca, NY” to the end

L298 – put 1984 in brackets

Reference 6 appears to be in a different format from the rest

Author Response

Reviewer 1

Comments and Suggestions for Authors

Animals 1837302

A non invasive sound technology to monitor rumen contractions

 This is a well-written technical note, only a few recommendations are made.

 The biggest concern is the lack of sample size (3) with no replication which is noted as a limiting factor. It is noted that the Curo MkII was used only once per cow for 20 min (L111-112), is this correct? Was there a reason why this was not extrapolated out for a longer period?

Authors: this time frame allowed us to record rumen movements and probe that the system works and can detect sounds and frequencies. Future work should have longer periods to assess other type of activities from the animal.

 I also understand that this is potentially commercially sensitive data (L289-291) and there was no funding provided (L280).

Authors: yes, we have sent a letter to the Editorial explaining this conflict of interest

 This manuscript can be recommended for publication as it is a technical note and is novel enough for consideration in spite of the lack of sample size.

Authors: thank you for your suggestions, all were considered

L250 states that the authors aim to move to an entire herd over a complete lactating period – this is a big jump from the present study which is only 3 cows for a total of 20 minutes

Authors: yes, after this technical note we are planning to apply for some funding – let us hope for the best outcomes…

A bit confused also of the range – it says 40 metres for Bluetooth but then “anywhere around the world”, what is the remote link and how does it work?

ANSWER: With regards to the range, the CURO units can both work as Bluetooth, and SIM card linked, where using Bluetooth the range is approx. 40 meters, but when coupled to a smart phone with a SIM card, the data can be sent anywhere around the globe.

CHANGE: None made, as we feel that lines 125-128 explain this. 

Minor comments:

L61-62 – a bit confused by this, the technology has the objective of automation and provide clues on ruminal movement dynamics but there is a challenge of monitoring systems? The monitoring systems should be an advantage, what is the challenge to obtain this real-time data?

Authors: for clarity, this has been re written, see line 61

L79 – change “To this end” to “In the present technical note,”

Authors: changed as suggested, see lines 80-81

L90 – how old was cow #1, was she a heifer?

Authors: this was changed to heifer, see line 92

L97 – change “dry” to “dry and pregnant” as per L225. Add “These two” before animals

Authors: this was changed as suggested, see lines 98

L100 – put grass silage before maize silage and put space between % and DM

Authors: this was changed as suggested, see lines 101

L101 – put space between % and DM

Authors: this was changed as suggested, see lines 102

L103 – after “kg of DM”, add “, and for dry cows based on 9.1 kg,”

Authors: this was changed as suggested, see lines 104-105

L104 – remove the final sentence

Authors: sentence was deleted as suggested

L106 – add “the” before solid black

Authors: due to copyrights, this figure was deleted, and some text was added instead, see lines 76-80

L128 – no need for capital w for world

Authors: changed as suggested, see line 125

Figure 3 – put the right panel in the middle as it flows better

Authors: changed as suggested

L155 – change “subject” to “animal”

Authors: changed as suggested, see line 150

L157 – were these licks and tail movements observed during the 20 minutes of experimentation?

Authors: yes, they were occasional during those 20 min

L170 – put comma after (234-239 Hz)

Authors: changed as suggested, see line 165

L173 – change “with-it” to “with it,”

Authors: changed as suggested, see line 167

L177-180 – these sentences starting from “Note that” can be removed as they were mentioned in the text

Authors: deleted as suggested

L184 - change “trace” to “panel”

Authors: changed as suggested, see line 176

L188 - change “trace” to “panel”

Authors: changed as suggested, see line 180

Figure 5 – it may benefit to label the two panels as “Cow #2” (Left 1) and “Cow #3” (Right 1), the left and right names are a bit confusing

Authors: changed as suggested, see lines 183-184

L222 – change “Co.” to “County”

Authors: changed as suggested, see line 214

L224 – is this paragraph comparing two non-lactating cows from different breeds? It may be worth mentioning Cow 3’s diet

Authors: changed as suggested, see lines 217-218

L252 – change “a dry cow” to “two Red Danish cows, one dry” as this compares the same breed at two different stages of lactation

Authors: changed as suggested, see lines 244-245

L265 – put in vivo in italics

Authors: changed as suggested, see line 258

L266 – put “and” between function and health and add comma after health

Authors: changed as suggested, see line 259

L269 – remove “the”

Authors: removed as suggested

References – only half of these show DOI and there are differences in “volume: pages” or “volume, pages”

Authors: all references were revised and changed to the journal’s format

Reference 2 – remove “New York State College Of, A., Life, S. & Department of M.” and move “Ithaca, NY” to the end

Authors: changed as suggested

L298 – put 1984 in brackets

Authors: according to the journal’s format, publication year does not have brackets

Reference 6 appears to be in a different format from the rest

Authors: changed as suggested

Additionally:

Authors: Regarding figure 1.

This figure has been removed and the following text added to the manuscript “Earlier, Wynn and colleagues [7] were able to record sound waves of ruminal contractions by standing in person alongside a ruminant with a stethoscope connected to a sound recorder. Wynn and colleagues [7] indicated ruminal contractions with solid black diamond symbols, from a recording made most likely soon after feeding, if not whilst the animal was being fed”. 

The remaining figures have now been re-numbered such that Figure 2 is now Figure 1 etc.

Reviewer 2 Report

The technical report showed that high-quality rumen sound waves were recorded by wireless devices to distinguish the rumen contraction of cows in different production states and physiological stages. This method has the advantages of lossless and real-time, and is more advanced than traditional methods. In terms of structure, the article is relatively complete and the discussion is substantial, but the results are not very significant. It is more a prospect for the future, and does not reflect obvious laws, such as how the acoustic signal corresponds to the physiological state or health state. In the conclusion part, the author points out that machine learning method can be used to process data, which is correct. You can refer to this literature: Zhang, M., Feng, H., Tomka, J., Polovka, M., Ma, R., & Zhang, X. (2021). Predicting of mutton sheep stress coupled with multi-environment sensing and supervised learning network in the transportation process. Computers and Electronics in Agriculture, 190, 106422. Finally, please improve the clarity of the picture, at least 300dpi

Author Response

Reviewer 2

Comments and Suggestions for Authors

The technical report showed that high-quality rumen sound waves were recorded by wireless devices to distinguish the rumen contraction of cows in different production states and physiological stages. This method has the advantages of lossless and real-time, and is more advanced than traditional methods. In terms of structure, the article is relatively complete and the discussion is substantial, but the results are not very significant. It is more a prospect for the future, and does not reflect obvious laws, such as how the acoustic signal corresponds to the physiological state or health state. In the conclusion part, the author points out that machine learning method can be used to process data, which is correct. You can refer to this literature: Zhang, M., Feng, H., Tomka, J., Polovka, M., Ma, R., & Zhang, X. (2021). Predicting of mutton sheep stress coupled with multi-environment sensing and supervised learning network in the transportation process. Computers and Electronics in Agriculture, 190, 106422. Finally, please improve the clarity of the picture, at least 300dpi

Authors: reference was added, and pictures were revised for definition

Additionally:

Authors: Regarding figure 1.

This figure has been removed and the following text added to the manuscript “Earlier, Wynn and colleagues [7] were able to record sound waves of ruminal contractions by standing in person alongside a ruminant with a stethoscope connected to a sound recorder. Wynn and colleagues [7] indicated ruminal contractions with solid black diamond symbols, from a recording made most likely soon after feeding, if not whilst the animal was being fed”. 

The remaining figures have now been re-numbered such that Figure 2 is now Figure 1 etc.

Round 2

Reviewer 1 Report

This manuscript is much improved. The minor comments below, once addressed, will mean that the manuscript is ready for publication.

Some very minor comments:

L79 – remove “with solid black diamond symbols,”

L80 – Wynn et al. [7] says that the recording was generally made before feeding

L103 – put space between % and DM

L144 – swap “middle” and “right”

L150 – change “subject” to “animal”

L152 – remove “indeed”

L158 – this should be Figure 1 (not Figure 3)

L167 – put comma after “with it”

L289 – remove “1984”

L291 – make 1984 bold

L307 – remove brackets from around 1984

Author Response

Reviewer 1

Comments and Suggestions for Authors

This manuscript is much improved. The minor comments below, once addressed, will mean that the manuscript is ready for publication.

Some very minor comments:

L79 – remove “with solid black diamond symbols,”

AUTHORS: removed as suggested, see line 79

L80 – Wynn et al. [7] says that the recording was generally made before feeding

AUTHORS: changed as suggested, see line 79

L103 – put space between % and DM

AUTHORS: changed as suggested, see line 103

L144 – swap “middle” and “right”

AUTHORS: changed as suggested, see line 144

L150 – change “subject” to “animal”

AUTHORS: changed as suggested, see line 150

L152 – remove “indeed”

AUTHORS: removed as suggested, see line 152

L158 – this should be Figure 1 (not Figure 3)

AUTHORS: changed as suggested, see line 158

L167 – put comma after “with it”

AUTHORS: changed as suggested, see line 167

L289 – remove “1984”

AUTHORS: changed as suggested, see line 289

L291 – make 1984 bold

AUTHORS: changed as suggested, see line 291

L307 – remove brackets from around 1984

AUTHORS: changed as suggested, see line 307
